# Causal Discovery from Subsampled Time Series with Proxy Variables

**Mingzhou Liu**[1,2]    **Xinwei Sun**[3*]    **Lingjing Hu**[4]    **Yizhou Wang**[1,2,5]

[1] School of Computer Science, Peking University
[2] Center on Frontiers of Computing Studies (CFCS), Peking University
[3] School of Data Science, Fudan University
[4] Yanjing Medical College, Capital Medical University
[5] Institute for Artificial Intelligence, Peking University

## Abstract

Inferring causal structures from time series data is the central interest of many scientific inquiries. A major barrier to such inference is the problem of subsampling, *i.e.*, the frequency of measurement is much lower than that of causal influence. To overcome this problem, numerous methods have been proposed, yet either was limited to the linear case or failed to achieve identifiability. In this paper, we propose a constraint-based algorithm that can identify the entire causal structure from subsampled time series, without any parametric constraint. Our observation is that the challenge of subsampling arises mainly from hidden variables at the unobserved time steps. Meanwhile, every hidden variable has an observed proxy, which is essentially itself at some observable time in the future, benefiting from the temporal structure. Based on these, we can leverage the proxies to remove the bias induced by the hidden variables and hence achieve identifiability. Following this intuition, we propose a proxy-based causal discovery algorithm. Our algorithm is nonparametric and can achieve full causal identification. Theoretical advantages are reflected in synthetic and real-world experiments. Our code is available at https://github.com/lmz123321/proxy_causal_discovery.

## 1   Introduction

Temporal systems are the primary subject of causal modeling in many sciences, such as pathology, neuroscience, and economics. For these systems, a common issue is the difficulty of collecting sufficiently refined timescale data. That is, we can only observe a *subsampled* version of the true causal interaction. This can cause serious problems for causal identification, since full observation (*i.e.*, causal sufficiency [1]) is believed to be an important condition for causal identification, and the violation of which (*e.g.*, the existence of hidden variables) can induce strong bias [2].

**Example** 1.1. Take the pathology study of Alzheimer's disease (AD) as an example. In AD, patients suffer from memory loss due to the atrophy of memory-related brain regions such as the Hippocampus [3]. To monitor such atrophy, the standard protocol is to perform a MRI examination on the brain every six months [4]. However, many studies have shown that the disease can progress much more rapidly than this [5, 6]. For this reason, when recovering the causal interactions between brain regions, those at the unobserved time steps constitute hidden variables and induce spurious edges in the causal graph, as shown in Figure 1 (b).

To resolve this problem, many attempts have been made to recover the correct causal relations from subsampled data. However, none of them achieved general identifiability results. For example, in

---

*Correspondence to sunxinwei@fudan.edu.cn

37th Conference on Neural Information Processing Systems (NeurIPS 2023).

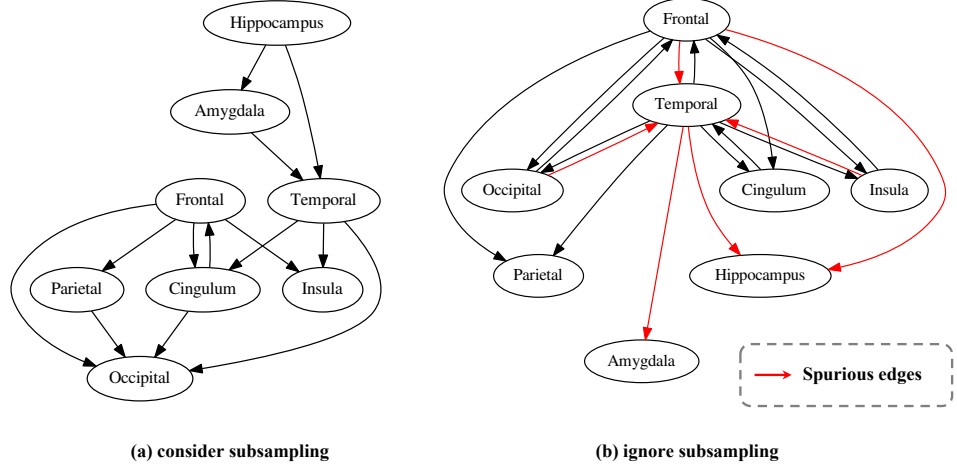

(a) consider subsampling        (b) ignore subsampling

Figure 1: Recovered causal pathways in Alzheimer's disease, with (a) our method that explicitly models the subsampling and (b) Dynotears [7] that neglects subsampling. Spurious edges that contradict clinical studies [8, 9] are marked red. Please refer to Section 5.2 and Figure 7 for details.

[10, 11, 12], identifiability was achieved only for linear data. As for nonlinear data, only a small part of the causal information (*i.e.*, an equivalence class) could be identified [13, 14, 15, 16]. Indeed, encountering such difficulties is no coincidence, considering the fact that there are many hidden variables in the system. Particularly, the hidden mediators prevent us from distinguishing the direct causation from possible mediation and hence only allow us to recover some ancestral information.

In this paper, we propose a constraint-based algorithm that can identify the entire causal structure from subsampled data, without any parametric assumption. Our departure here is that in time series, every hidden variable has an observed descendent, which is essentially itself at some observable time in the future[1]. Therefore, we can use the observed descendent as its *proxy variable* [17] and adjust for the bias induced by it. For this purpose, we first represent the ancestral information, which is composed of both direct causation and hidden mediation, with the Maximal Ancestral Graph (MAG) [18]. Then, to distinguish causation from mediation, we use the observed descendants of the hidden mediators as their proxy variables and adjust for their induced mediation bias. We show that our strategy can achieve a complete identification of the causal relations entailed in the time series. It is essentially nonparametric and only relies on the smoothness and completeness of the structural equation to conduct valid proxy-based adjustment [19].

Back to the AD example, as shown in Figure 1, our method can recover causal pathways that align well with existing clinical studies [8, 9], while compared baselines fail to do so.

Our contributions are summarized below:

1. We establish nonparametric causal identification in subsampled time series.

2. We propose an algorithm to practically recover the causal graph.

3. We achieve more accurate causal identification than others on synthetic and real data.

The rest of the paper is organized as follows. First, in Section 2, we introduce the problem setting, basic assumptions, and related literature needed to understand our work. Second, in Section 3, we establish theories on structural identifiability. Then, based on these theories, in Section 4, we introduce the proposed causal discovery algorithm and verify its effectiveness in Section 5. Finally, we conclude the paper and discuss future works in Section 6.

---

[1]For example, the hidden variable $X(t_u)$ at the unobserved time step $t_u$ with $X(t_u) \rightarrow X(t_u + 1) \rightarrow X(t_u + 2) \rightarrow \cdots \rightarrow X(t_o)$ has an observable descendent $X(t_o)$ at some observable time $t_o$ in the future.

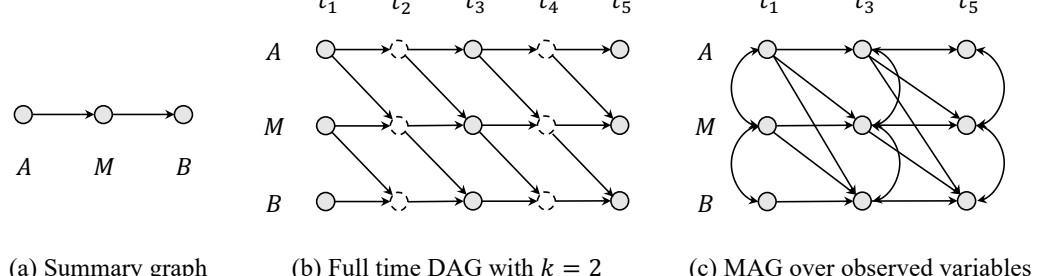

| (a) Summary graph | (b) Full time DAG with $k = 2$ | (c) MAG over observed variables |

Figure 2: Graph terminologies: (a) Summary graph. (b) Full time DAG with subsampling factor $k = 2$. Variables at observed time steps $t_1, t_3, t_5$ are marked gray, while those at unobserved time steps $t_2, t_4$ are dashed. (c) MAG over observed variables. The edges $A(t_1) \rightarrow M(t_3)$ and $A(t_3) \leftrightarrow M(t_3)$ are (*resp.*) induced by the inducing paths $A(t_1) \rightarrow A(t_2) \rightarrow M(t_3)$ and $A(t_3) \leftarrow A(t_2) \rightarrow M(t_3)$ in the full time DAG. Please refer to the appendix for detailed explanations.

## 2 Preliminary

We start by introducing the causal framework and related literature needed to understand our work.

**Time series model.** Let $\mathbf{X}(t) := [X_1(t), ..., X_d(t)]$ be a multivariate time series with $d$ variables defined at discrete time steps $t = 1, ..., T$. We assume the data is generated by a first-order structural vector autoregression (SVAR) process [16]:

$$X_i(t) = f_i(\mathbf{PA}_i(t-1), N_i), \tag{1}$$

where $f_i$ is the structural function, $\mathbf{PA}_i$ is the parent of $X_i$, and $N_i$ is the exogenous noise.

Implicit in (1) is the assumptions that cause precedes effect, and that causation is invariant to time, *i.e.*, the structural function $f_i$ and the causal parents $\mathbf{PA}_i$ keeps unchanged across time steps. These assumptions carry the fundamental beliefs of temporal precedence and stability on causality [1], therefore are widely adopted by existing works [10, 11, 12, 14, 16].

**Subsampling.** The subsampling problem means that model (1) can be only observed every $k$ steps [10, 12, 16]. That is, we can only observe $\mathbf{X}(1), \mathbf{X}(k+1), ..., \mathbf{X}(\lfloor \frac{T}{k} \rfloor k + 1)$.

**Graph terminologies.** Causal relations entailed in model (1) can be represented by causal graphs (Figure 2). We first introduce the *full time Directed Acyclic Graph (DAG)* [20], which provides a complete description of the dynamics in the system.

**Definition 2.1** (Full time DAG). Let $G := (\mathbf{V}, \mathbf{E})$ be the associated full time DAG of model (1). The vertex set $\mathbf{V} := \{\mathbf{X}(t)\}_{t=1}^T$, the edge set $\mathbf{E}$ contains $X_i(t-1) \rightarrow X_j(t)$ iff $X_i \in \mathbf{PA}_j$.

We assume the full time DAG is Markovian and faithful to the joint distribution $\mathbb{P}(\mathbf{X}(1), ..., \mathbf{X}(T))$.

**Assumption 2.2** (Markovian and faithfulness). For disjoint vertex sets $\mathbf{A}, \mathbf{B}, \mathbf{Z} \subseteq \mathbf{V}$, $\mathbf{A} \perp\!\!\!\perp \mathbf{B}|\mathbf{Z} \Leftrightarrow \mathbf{A} \perp\!\!\!\perp_G \mathbf{B}|\mathbf{Z}$, where $\perp\!\!\!\perp_G$ denotes $d$-separation in $G$.

In practice, it is often sufficient to know the causal relations between time series as a whole, without knowing precisely the relations between time instants [21]. In this regard, we can summarize the causal relations with the *summary graph* [20].

**Definition 2.3** (Summary graph). Let $G := (\mathbf{V}, \mathbf{E})$ be the associated summary graph of model (1). The vertex set $\mathbf{V} := \mathbf{X}$, the edge set $\mathbf{E}$ contains $X_i \rightarrow X_j (i \neq j)$ iff $X_i \in \mathbf{PA}_j$.

In this paper, our goal is to recover the summary graph, given data observed at $t = 1, k+1, ..., \lfloor \frac{T}{k} \rfloor k + 1$.

For this purpose, we need a structure to represent the (marginal) causal relations between observed variables and hence link the observation distribution to the summary graph. Here, we use the *Maximal Ancestral Graph (MAG)* [22], since it can represent casual relations when unobserved variables exist.

Specifically, for the variable set $\mathbf{V} := \{\mathbf{X}(t)\}_{t=1}^T$, let $\mathbf{O} := \{\mathbf{X}(1), ..., \mathbf{X}(\lfloor \frac{T}{k} \rfloor k + 1)\}$ be the observed subset and $\mathbf{L} := \mathbf{V} \backslash \mathbf{O}$ be the unobserved subset. Let $\mathbf{An}(A)$ be the ancestor set of $A$. Then, given any full time DAG $G$ over $\mathbf{V}$, the corresponding MAG $M_G$ over $\mathbf{O}$ is defined as follows:

**Definition 2.4** (MAG). In the MAG $M_G$, two vertices $A, B \in \mathbf{O}$ are adjacent iff there is an inducing path. [2] relative to $\mathbf{L}$ between them in $G$. Edge orientation is:

1. $A \rightarrow B$ in $M_G$ if $A \in \mathbf{An}(B)$ in $G$;

2. $A \leftarrow B$ in $M_G$ if $B \in \mathbf{An}(A)$ in $G$;

3. $A \leftrightarrow B$ in $M_G$ if $A \notin \mathbf{An}(B)$ and $B \notin \mathbf{An}(A)$ in $G$.

*Remark* 2.5. In the MAG, the directed edge $A \rightarrow B$ means $A$ is the ancestor of $B$. The bidirected edge $A \leftrightarrow B$ means there is an unobserved confounder $U$ between $A$ and $B$.

**Proximal causal discovery.** Recent works [23, 19] found that one could use the descendant (*i.e.*, proxy) of an unobserved variable $M$ to differentiate direct causation $A \rightarrow B$ from hidden mediation $A \rightarrow M \rightarrow B$. Specifically, suppose that:

**Assumption 2.6** (Smoothness). For each causal pair $A \rightarrow B$, the map $a \mapsto \mathbb{P}(B|A = a)$ is Lipschitz with respect to the Total Variation (TV) distance, that is, there exists a constant $L$ such that:

$$\mathrm{TV}(\mathbb{P}(B|A = a), \mathbb{P}(B|A = a')) \leq L|a - a'|.$$

**Assumption 2.7** (Completeness). For each causal pair $A \rightarrow B$, the conditional distribution $\mathbb{P}(B|A)$ is complete, that is, for any function $g$, we have:

$$E\{g(b)|a\} = 0 \text{ almost surely if and only if } g(b) = 0 \text{ almost surely.}$$

*Remark* 2.8. Roughly, Assumption 2.6 relies on the continuity of the structural function $f_i$ and Assumption 2.7 requires that the characterization function of the exogenous noise $N_i$ is non-zero. Please refer to [19] for a detailed discussion.

Under the above assumptions, [19] showed that:

**Theorem 2.9** ([19]). *Suppose that $A$ and $B$ are mediated by an unobserved variable $M$, and that $M$ has a proxy variable $M'$ satisfying $A \perp\!\!\!\perp_G M'|M$. Then, we can identify whether $A \rightarrow B$ by testing whether $pr(b|a) \sim pr(m'|a)$ has a linear relation. If the linear relation exists, then $A \nrightarrow B$, otherwise, we have $A \rightarrow B$.*

*Remark* 2.10. The linear relation can be tested by checking the least square residual of regressing $pr(b|a)$ on $pr(m'|a)$. Please refer to [19] for details.

To ensure the existence of the proxy variable, we require the following self causation assumption, which states that each variable $X_i(t)$ is influenced by its past $X_i(t-1)$.

**Assumption 2.11** (Self causation). In model (1), $X_i \in \mathbf{PA}_i$ for each $i = 1, ..., d$.

*Remark* 2.12. In time series statistics, this is also known as the autocorrelation assumption [24, 25].

## 3 Structural identifiability

In this section, we establish the identifiability of the summary graph. For this purpose, we connect the observational distribution to the summary graph with the bridge of the MAG.

Specifically, our analysis consists of three progressive results: Proposition 3.1, Proposition 3.3, and Theorem 3.5. First, in Proposition 3.1, we prove that the MAG can be identified from observational data. Then, in Proposition 3.3, we show that the identified MAG *almost uniquely* reflects the structure of the summary graph, in the sense that only edges connecting vertices and their ancestors cannot be determined due to hidden mediators. Finally, in Theorem 3.5, we identify these underdetermined edges with the proxies of the hidden mediators and therefore identify the whole summary graph.

---

[2]A path $p$ between $A, B \in \mathbf{O}$ is an inducing path relative to $\mathbf{L}$ if every non-endpoint vertex on $p$ is either in $\mathbf{L}$ or a collider, and every collider on $p$ is an ancestor of either $A$ or $B$.

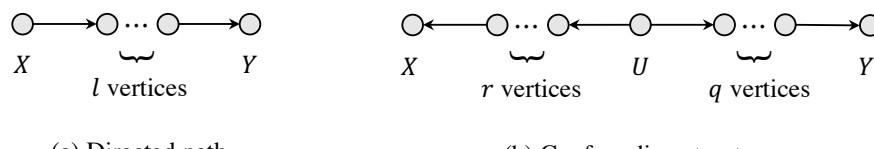

(a) Directed path                  (b) Confounding structure

Figure 3: Illustration of Definition 3.2. (a) A directed path from $A$ to $B$ with length $l$. (b) A confounding structure between $A$ and $B$ with lengths $(r, q)$.

Next, we first introduce Proposition 3.1, which shows that the MAG is identifiable.

**Proposition 3.1** (Identifiability of the MAG). *Assuming model (1) and Assumption 2.2, then the MAG over the observed variable set $\mathbf{O}$ is identifiable, i.e., its skeleton and edge orientations can be uniquely derived from the joint distribution $\mathbb{P}(\mathbf{O})$.*

Since the identifiability of the MAG's skeleton under the faithfulness assumption is a well-known result [18], we focus on explaining the identification of edge orientations. Specifically, we will divide the edges into two classes: the instantaneous edges and the lagged edges, and discuss their orientations respectively.

The *instantaneous edge*, e.g., $A(t_3) \leftrightarrow M(t_3)$, is an edge that connects two vertices at the same time. Since we assume cause must precede effect, the instantaneous edge does not represent causation but latent confounding, therefore is bidirected according to Definition 2.4. For example, in Figure 2 (c), the instantaneous edge $A(t_3) \leftrightarrow M(t_3)$ represents the latent confounding $A(t_3) \leftarrow A(t_2) \rightarrow M(t_3)$ between $A(t_3)$ and $M(t_3)$. On the other hand, the *lagged edge*, e.g., $A(t_1) \rightarrow M(t_3)$, is an edge that connects two vertices at different time. The lagged edge represents ancestral information and hence is directed ($\rightarrow$) from the past to the future.

To connect the identified MAG to the summary graph, we first define the following graph structures, examples of which are shown in Figure 3:

**Definition 3.2** (Graph structures). In the summary graph,

1. A *directed path* $p_{AB}$ from $A$ to $B$ with length $l$ is a sequence of distinct vertices $A, V_1, ..., V_l, B$ where each vertex points to its successor. Two directed paths $p_{AB_1}, p_{AB_2}$ are called *disjoint* if they do not share any non-startpoint vertex.

2. A *confounding structure* $c_{AB}$ between $A$ and $B$ with lengths $(r, q)$ consists of a vertex $U$, a directed path $p_{UA}$ from $U$ to $A$ with length $r$, and a directed path $p_{UB}$ from $U$ to $B$ with length $q$, where $p_{UA}$ and $p_{UB}$ are disjoint.

In the following, we explain the relationship between the identified MAG and the summary graph. In particular, we will discuss what the directed and bidirected edges in the MAG (*resp.*) imply about the summary graph.

According to Definition 2.4, the *directed edge* $A(t) \rightarrow B(t + k)$ in the MAG means $A$ is the ancestor of $B$. Therefore, in the summary graph, there is either $A \rightarrow B$ or a directed path from $A$ to $B$. On the other hand, the *bidirected edge* $A(t + k) \leftrightarrow B(t + k)$ in the MAG means there is a latent confounder between them, which can be either $A(t + 1)$, i.e., $A(t + k) \leftarrow \cdots \leftarrow A(t + 1) \rightarrow \cdots \rightarrow B(t + k)$, or a third variable $U(t + 1)$, i.e., $A(t + k) \leftarrow \cdots \leftarrow U(t + 1) \rightarrow \cdots \rightarrow B(t + k)$. Hence, in the summary graph, there is either a directed path from $A$ to $B$ or a confounding structure between them.

Summarizing the above observations, we have the following proposition:

**Proposition 3.3** (MAG to summary graph). *If there are $A(t) \rightarrow B(t + k)$ and $A(t + k) \leftrightarrow B(t + k)$ in the MAG, then, in the summary graph, there is either*

   *1. $A \rightarrow B$,   2. a directed path from $A$ to $B$ with length $l \leq k - 2$, or*

   *3. a directed path from $A$ to $B$ with length $l = k - 1$ and a confounding structure between them with lengths $(r \leq k - 2, q \leq k - 2)$.*

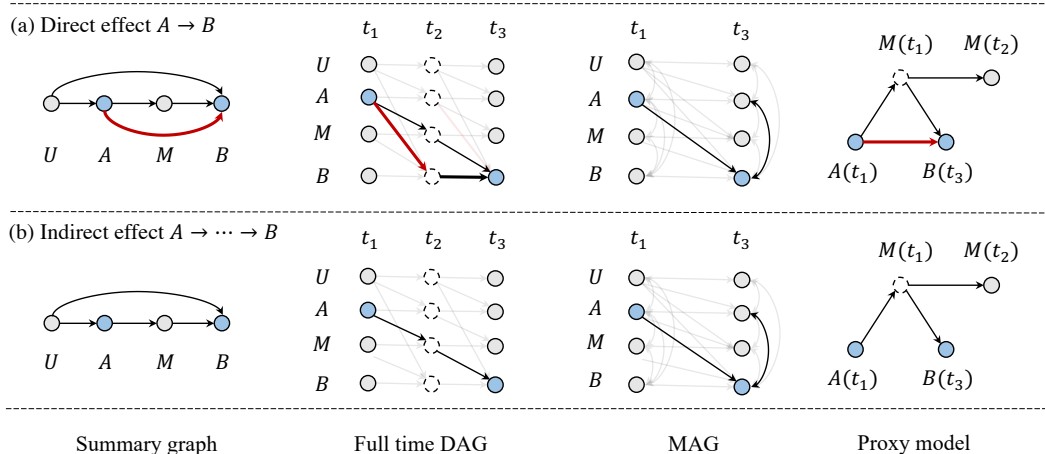

Figure 4: Distinguishing the direct effect ($A \to B$) from the indirect one ($A \to \cdots \to B$) with proxy variables. Note that though (a) and (b) have the same MAG, only the indirect effect $A(t_1) \to M(t_2) \to B(t_3)$ in (b) can be $d$-separated by $M(t_2)$. To test the $d$-separation, we can use $M(t_3)$ as the proxy variable.

*Remark* 3.4. In Proposition 3.3, the maximum length of the directed path is $k - 1$, this is because any path longer than this can not be an inducing path and therefore can not induce $A(t) \to B(t + k)$ in the MAG. Please refer to the appendix for details.

Proposition 3.3 provides a necessary condition for having $A \to B$ in the summary graph, *i.e.*, there are both $A(t) \to B(t + k)$ and $A(t + k) \leftrightarrow B(t + k)$ in the MAG. It also inspires us that, to make the condition sufficient, we need to distinguish the direct effect ($A \to B$) from the indirect one ($A \to \cdots \to B$).

For this purpose, we propose to use the proximal causal discovery method. Before preceding any technical detail, we first brief our idea below. The indirect effect, unlike the direct one, relies on mediation and therefore can be $d$-separated by the (unobserved) mediator. To test the $d$-separation, we can use the observed proxy of the mediator, thanks to the self causation assumption in Assumption 2.11. For example, in Figure 4 (b), the indirect effect $A(t_1) \to M(t_2) \to B(t_3)$ from $A$ to $B$ can be $d$-separated by the mediator $M(t_2)$, who has an observed proxy $M(t_3)$.

For the general case, the separation set is a bit more complicated to ensure all paths except the one representing direct effect are $d$-separated. Specifically, for two vertices $A(t)$ and $B(t + k)$, the separation set is the union of two sets $\mathbf{M}(t + 1) \cup \mathbf{S}(t)$. The set $\mathbf{M}(t + 1)$ contains possible mediators, namely $A(t + 1)$ and any vertex $M_i(t + 1)$ ($M_i \neq B$) such that $A(t) \to M_i(t + k)$ in the MAG and $M_i$ is not $B$'s descendant[3]. The set $\mathbf{S}(t)$ is used to $d$-separate possible back-door paths between $A(t)$ and $B(t + k)$, it contains any vertex $S_i(t)$ ($S_i \neq A$) such that $S_i(t) \to B(t + k)$ or $S_i(t) \to M_j(t + k)$ for some $M_j \in \mathbf{M}$ in the MAG.

Equipped with the separation set, we then have the following identifiability result:

**Theorem 3.5** (Identifiability of the summary graph). *Assuming model (1), Assumption 2.2, and Assumptions 2.6, 2.7, 2.11, then the summary graph is identifiable. Specifically,*

1. *There is $A \to B$ in the summary graph iff there are $A(t) \to B(t + k), A(t + k) \leftrightarrow B(t + k)$ in the MAG, and the set $\mathbf{M}(t + 1) \cup \mathbf{S}(t)$ is not sufficient to d-separate $A(t), B(t + k)$ in the full time DAG.*

2. *The condition "the set $\mathbf{M}(t + 1) \cup \mathbf{S}(t)$ is not sufficient to d-separate $A(t), B(t + k)$ in the full time DAG" can be tested by the proxy variable $\mathbf{M}(t + k)$ of the unobserved set $\mathbf{M}(t + 1)$.*

---

[3]This can be justified from the MAG. Specifically, if $M_i$ is the descendant of $B$, then there is $B(t) \to V_1(t + k), V_1(t) \to V_2(t + k), ..., V_l(t) \to M_i(t + k)$ in the MAG.

Theorem 3.5 provides a necessary and sufficient condition for having $A \to B$ in the summary graph. This condition is testable with the proxy $\mathbf{M}(t+k)$ of the unobserved set $\mathbf{M}(t+1)$ ($\mathbf{S}(t)$ is observable). The validity of such proxying, *i.e.*, $A(t) \perp\!\!\!\perp_G \mathbf{M}(t+k)|\mathbf{M}(t+1), \mathbf{S}(t)$ is proved in the appendix.

## 4 Discovery algorithm

Equipped with the identifiability results in Proposition 3.1 and Theorem 3.5, we can practically discover the summary graph with Algorithm 1. Specifically, given the observed data points, Algorithm 1 first recovers the MAG according to Proposition 3.1. Then, for each pair of vertices in the summary graph, it first uses the necessary condition of "having $A(t) \to B(t+k)$ and $A(t+k) \leftrightarrow B(t+k)$ in the MAG" to construct underdetermined edges. Finally, among these underdetermined edges, it uses the proxy variable approach to identify direct causation and remove the indirect one, according to Theorem 3.5.

Below, we first introduce the *Partially Determined Directed Acyclic Graph (PD-DAG)*, which will be used to represent the intermediate result of the algorithm.

**Definition 4.1** (PD-DAG). A PD-DAG is a DAG with two kinds of directed edges: solid ($\to$) and dashed ($\dashrightarrow$). A solid edge represents a determined causal relation, while a dashed one means the causal relation is still underdetermined. We use a meta-symbol, asterisk ($\ast\!\to$), to denote any of the two edges. In a PD-DAG $G = (\mathbf{V}, \mathbf{E})$, the vertex $A$ points to the vertex $B$ if the edge $A \ast\!\to B \in \mathbf{E}$. The definition of the directed path and confounding structure are the same as in the summary graph.

---

**Algorithm 1:** Discover the summary graph

**Data:** Data observed at $t = 1, k+1, ..., \lfloor \frac{T}{k} \rfloor k + 1$
**Result:** The summary graph

1 Construct the skeleton of the MAG with [26], orient edges according to Proposition 3.1;
2 For every vertex pair $A, B$, set $A \ast\!\to B$ if there are $A(t) \to B(t+k)$ and $A(t+k) \leftrightarrow B(t+k)$
   in the MAG. Called the resulted PD-DAG $G_0$ ;       /* Proposition 3.3 */
3 Set $G = G_0$ and iteratively execute:

    (a) For every $A \ast\!\to B$, check

        (i) a directed path from $A$ to $B$ with length $l \leq k-2$;
        (ii) a directed path from $A$ to $B$ with length $l \leq k-1$ and a confounding structure between
            $A$ and $B$ with lengths $(r \leq k-2, q \leq k-2)$;

       If $\exists$ (i) or (ii), pass. Otherwise, set $A \to B$ ;       /* Proposition 3.3 */

    (b) Randomly pick a dashed edge $A \ast\!\to B$, check whether $A(t) \perp\!\!\!\perp_G B(t+k)|\mathbf{M}(t+1), \mathbf{S}(t)$;
       If the $d$-separation holds, remove $A \ast\!\to B$. Otherwise, set $A \to B$ ;  /* Theorem 3.5 */

until all edges in $G$ are solid ones; **return** $G$

---

*Remark* 4.2. Algorithm 1 can be generalized to cases where the subsampling factor $k$ is unknown, by skipping the step (a) below line 3.

## 5 Experiment

In this section, we evaluate our method on synthetic data and a real-world application, *i.e.*, discovering causal pathways in Alzheimer's disease.

**Compared baselines.** *Methods that account for subsampling*: 1. SVAR-FCI [16] that extended the FCI algorithm to time series and recovered a MAG over observed variables; 2. NG-EM [10] that achieved identifiability with linear non-Gaussianity and used Expectation Maximization (EM) for estimation. *Methods that neglect subsampling*: 1. Dynotears [7] that extended the score-based Notears [27] algorithm to the time series; 2. PC-GCE [21] that modified the PC algorithm for temporal data with a new information-theoretic conditional independence (CI) test.

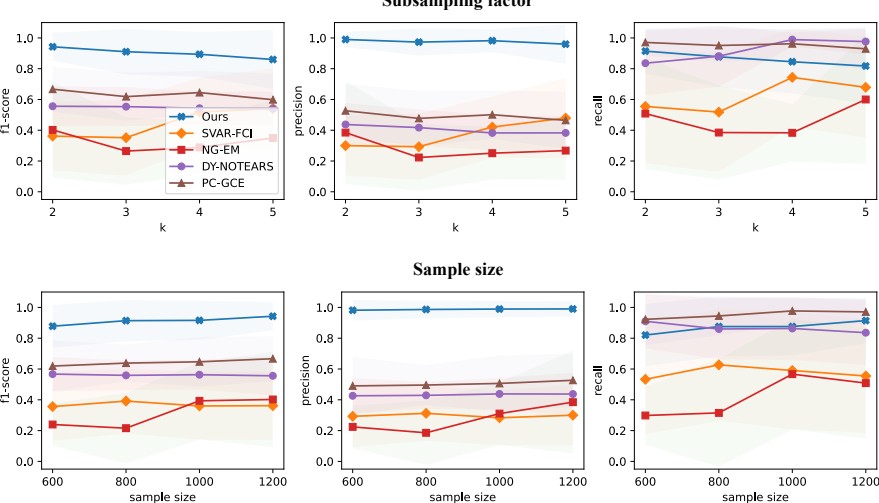

Figure 5: Performance of our method and baselines under different subsampling factors (upper row) and sample sizes (lower row).

**Metrics.** We use the $F_1$-score, precision, and recall, where precision and recall (*resp.*) measure the accuracy and completeness of identified causal edges, and $F_1 := 2 \cdot \frac{\text{precision} \cdot \text{recall}}{\text{precision} + \text{recall}}$.

**Implementation details.** The significance level is set to $0.05$. For the MAG recovery, we use the FCI algorithm implemented in the $\mathrm{causallearn}$ package[4]. Temporal constraints, *e.g.*, causal precedence, time invariance, and self causation are added as the background knowledge.

## 5.1 Synthetic study

**Data generation.** We generate radnom summary graphs with the Erdos-Renyi model [28], where the vertex number is set to $5$, the probability of each edge is set to $0.3$. For each graph, we generate temporal data with the structural equation $X_i(t) = \sum_{j \in \mathbf{PA}_i} f_{ij}(X_j(t-1)) + N_i$, where the function $f_{ij}$ is randomly chosen from $\{linear, sin, tanh, sqrt\}$, the exogenous noise $N_i$ is randomly sampled from $\{uniform, gauss., exp., gamma\}$. We consider different subsampling factors $k = \{2, 3, 4, 5\}$ and sample sizes $n = \{600, 800, 1000, 1200\}$. For each setting[5], we replicate over $100$ random seeds.

**Comparison with baselines.** Figure 5 shows the performance of our method and baselines under different subsampling factors (upper row) and sample sizes (lower row). As we can see, our method significantly outperforms the others in all settings. Specifically, compared with methods that account for subsampling (SVAR-FCI and NG-EM), our method achieves both higher precision and recall, indicating less detection error and missing edges. This advantage can be attributed to the fact that our method enjoys an identifiability guarantee (*v.s.* SVAR-FCI) and meanwhile requires no parametric assumption on the structural model (*v.s.* NG-EM). Compared with methods that ignore subsampling (Dynotears and PC-GCE), our method achieves higher $F_1$-score, precision, and is comparable in recall. This result shows that our method can effectively reduce the spurious detection induced by unobserved time steps. Besides, we can observe that Dynotears and PC-GCE slightly outperform SVAR-FCI and NG-EM, which again demonstrates the necessity of establishing nonparametric identification in subsampling problems.

**Intermediate results.** We evaluate the intermediate results of Algorithm 1 (MAG, edges identified by proxies, summary graph) and report the results in Figure 6. As shown, both the recovered MAG and edges identified by proxies have high accuracy under moderate subsampling factors and sample sizes, hence explaining the effectiveness of our algorithm in recovering the summary graph. Meanwhile, we can also observe that the performance slightly decreases when the subsampling factor $k$ is large. This

---

[4] $\mathrm{https://github.com/py\text{-}why/causal\text{-}learn}$

[5] We also consider different graph scales $d = \{5, 15, 25, 35, 45\}$. Please refer to the appendix for details.

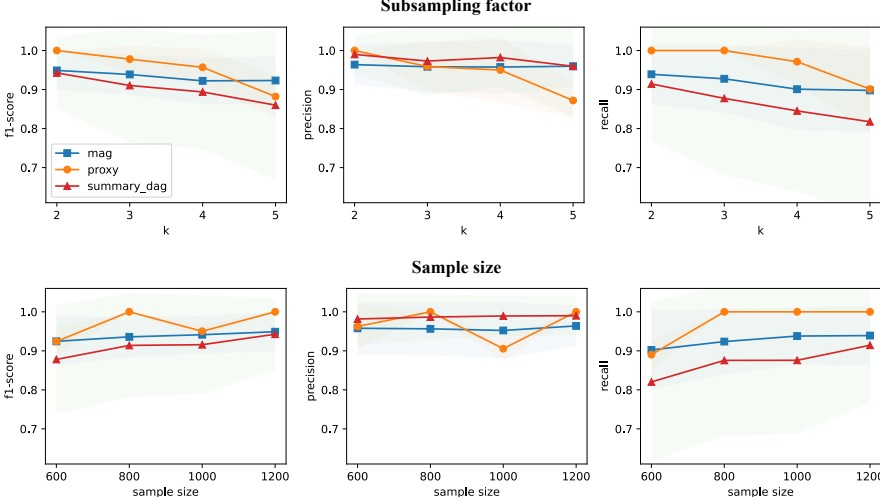

Figure 6: Evaluation of intermediate results. We report $F_1$-score, precision, and recall of the recovered MAG, edges identified by proxies, and the recovered summary graph.

is explained as follows. When $k$ is large, there are many unobserved steps between two observations, which weakens the correlation pattern in data and therefore breaks our faithfulness assumption.

## 5.2 Discovering causal pathways in Alzheimer's disease

**Background.** Alzheimer's disease (AD) is one of the most common neuro-degenerative diseases. In AD, patients suffer from memory loss due to atrophy of memory-related brain regions, such as the Hippocampus [3] and Temporal lobe [29]. A widely accepted explanation for such atrophy is: the disease releases toxic proteins, *e.g.*, $A\beta$ [30] and $tau$ [31]; along anatomical pathways, these proteins spread from one brain region to another, eventually leading to atrophy of the whole brain [32]. Recovering these anatomical pathways, *i.e.*, underlying causal mechanisms, will benefit the understanding of AD pathology and inspire potential treatment methods.

**Dataset and preprocessing.** We consider the Alzheimer's Disease Neuroimaging Initiative (ADNI) dataset [4], in which the imaging data is acquired from structural Magnetic Resonance Imaging (sMRI) scans. We apply the Dartel VBM [33] for preprocessing and the Statistical Parametric Mapping (SPM) [34] for segmenting brain regions. Then, we implement the Automatic Anatomical Labeling (AAL) atlas [35] to partition the whole brain into 90 regions. In total, we use $n = 558$ subjects with baseline and month-6 follow-up visits enrolled in ADNI-GO/1/2/3 periods.

**Results.** Figure 7 shows the recovered summary graph over 90 brain regions. For illustration, in Figure 1 (a), we further group the brain regions into 8 meta-regions according to their anatomical structures. We can observe that **i)** most causal pathways between meta-regions are unidirectional; **ii)** the identified sources of atrophy are the Hippocampus, Amygdala, and Temporal lobe, which are all early-degenerate regions [3, 29]; **iii)** the topology order induced from our result, *i.e.*, $\{Hip., Amy., Tem., Cin., Ins., Fro., Par., Occ.\}$ coincides with the temporal degeneration order found in [8, 9]. These results constitute an important finding that can be supported by existing studies: the atrophy (releasing of toxic proteins) is sourced from the Hippocampus and gradually propagates to other brain structures along certain anatomical pathways [32, 36].

In contrast, in Figure 1 (b), the causal relations recovered by the Dynotears baseline[6] are less clear. For example, most of the identified interactions are bidirectional, which may be due to the spurious edges induced by subsampling. Besides, the identified atrophy source is the Frontal lobe, with AD-related regions identified as outcomes, which is inconsistent with clinical studies. These results, from another perspective, show the importance of modeling subsampling in time series causal discovery.

---

[6]Please refer to the appendix for results of the other baselines.

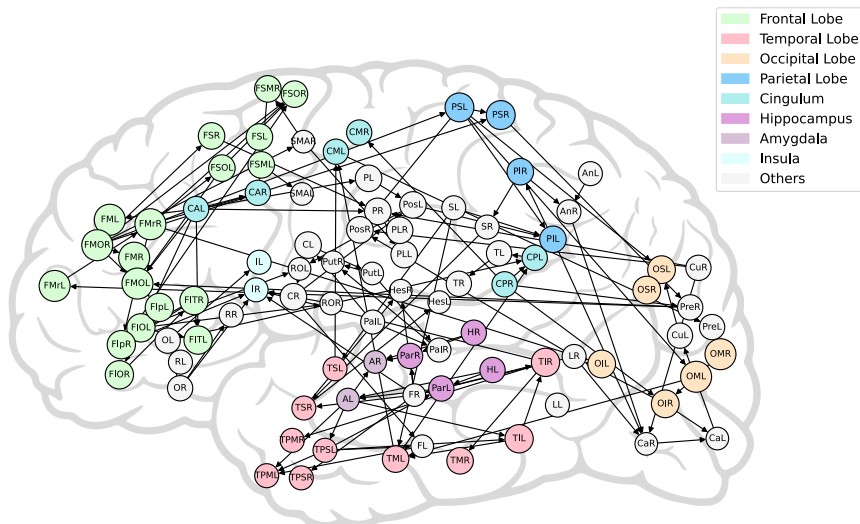

Figure 7: Recovered summary graph in Alzheimer's disease.

# 6    Conclusion

In this paper, we propose a causal discovery algorithm for subsampled time series. Our method leverages the recent progress of proximal causal discovery and can achieve complete identifiability without any parametric assumption. The proposed algorithm can outperform baselines on synthetic data and recover reasonable causal pathways in Alzheimer's disease.

**Limitation and future work.** Our method relies on an accurate test of the conditional independence (CI) and therefore may suffer from low recall when the CI patterns in data are weak. To solve this problem, we will investigate theories on high-efficiency CI tests and pursue a dedicated solution.

# Acknowledgments

This work was supported by National Key R&D Program of China (2022ZD0114900).

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
