# OpenReview forum: "Causal Discovery from Subsampled Time Series with Proxy Variables"
_NeurIPS.cc/2023/Conference — NeurIPS 2023 poster_

### Official Review · Reviewer_3AR7 · 2023-06-20

**Soundness:** 4 excellent
**Presentation:** 3 good
**Contribution:** 3 good
**Rating:** 7
**Confidence:** 2

**Summary:**

The authors suggest a method of causal discovery for multivariate time series under the regime when the data is being sampled at constant skips in the time dimension. Under mild assumptions, they prove that their method works asymptotically.

**Strengths:**

The paper's contribution is clear, the methods are interesting and novel, and the subject (causal inference in multivariate time series) is of interest to many. Other than some places I will mention, the explanations are clear.

**Weaknesses:**

I would like to see a little more discussion on previous work. I see citations about work that inspired the current one, for example [17, 19, 25] on using descendants of an unobserved variable to differentiate direct causation from hidden mediation, but I do not see citations for competing methods for causality under the regime of sampling on the time dimension. It would be good to be explicit about what the current state of the art is in that direction, and how the current work compares. You mention them in Section 5 (SVAR-FCI, NG-EM, Dynotears, PC-GCE) but I'd like to see a bigger discussion on the differences. Specifically, it would be great if we had a basic example in mind where "naive" approaches clearly fail, and make us understand how the proposed method avoids the issue intuitively.

**Questions:**

1. I feel like Definition 2.3 can be phrased more simply. The way you wrote it makes it look circular.
2. I am a little confused by the figures. In Figure 2.c., I don't understand why there wouldn't be an arrow from A(t_3) to M(t_5). Or perhaps the faded arrows count? Why is there a difference between faded and non-faded arrows?
3. In Figure 3, where are some vertices faded and some not? Is there a difference? It looks like it is meant that faded means unobserved, but I don't think that's what you meant.
4. Put Remark 4.2 outside of the algorithm.
5. Line 224: random is misspelled.


**Limitations:**

None.

---

> ### Author Rebuttal · Authors · 2023-08-09
>
> Thank you for the positive assessment and valuable suggestions on our paper. We will modify the manuscript accordingly.
>
> **About related works.** The existing methods have been discussed in lines 30-36. To summarize, identifiability is only achieved in linear data. As for nonlinear data, only a small part of the causal information (i.e., an equivalence class) can be identified.
>
> **About the example.** Please refer to Fig. 4 for the illustration example. In particular, naive approaches can not distinguish between the direct effect $A\to B$ and the indirect one $A\to M\to B$, due to the latent mediator $M(t_2)$. Our method solves this problem by using $M(t_3)$ as the proxy variable of $M(t_2)$.
>
> **About figures.** For both Fig. 2c and Fig. 3, there is no difference between faded and non-faded arrows/vertices. We will modify these figures as suggested.

---

> > ### Comment · Reviewer_3AR7 · 2023-08-17
> >
> > Thanks, I will keep the rating.

---

### Official Review · Reviewer_Y9xG · 2023-06-29

**Soundness:** 3 good
**Presentation:** 3 good
**Contribution:** 3 good
**Rating:** 6
**Confidence:** 4

**Summary:**

# Summary
In this paper, the authors address the problem of inferring causal structures from subsampled time series data, where the frequency of measurement is much lower than that of causal influence. This presents challenges in identifying the causal structure, as hidden variables at unobserved time steps can induce bias. Existing methods that tackle this problem are limited to linear cases or fail to achieve identifiability.

The main contribution of this paper is a constraint-based algorithm that can identify the entire causal structure from subsampled time series without any parametric constraints. The authors propose a proxy-based causal discovery algorithm that leverages the temporal structure of time series data to remove the bias induced by hidden variables. The algorithm is nonparametric and can achieve full causal identification. Specifically, the author leverages the proxy variables to test the edge directions of the summary DAG from the uniquely identified MAG.

The authors demonstrate the theoretical advantages of their method and provide experiments on both synthetic and real-world data, showcasing improved performance over existing methods.

**Strengths:**

# Originality

The paper presents a novel approach to causal discovery in subsampled time series data by proposing a constraint-based algorithm that leverages proxy variables to handle the challenges posed by hidden variables at unobserved time steps. The originality of the method lies in its ability to identify the entire causal structure without any parametric constraints, setting it apart from existing methods that are limited to linear cases or fail to achieve full identifiability. The authors draw inspiration from the recent progress in proximal causal discovery and adapt it to the subsampled time series setting.

# Quality
The quality of the paper is high, as it presents a well-formulated methodology with solid theoretical foundations. The authors provide rigorous proofs for their proposed algorithm's identifiability properties, ensuring that the algorithm is grounded in strong theoretical underpinnings. Although I have some questions regarding its generality and required assumptions, I will elaborate about them later.

As for the experiments, the author conduct 1 synthetic and 1 real-world experiment, which demonstrates its effectiveness against the baselines.

# Clarity

The paper is well-written and clear in its presentation. The methodogy is clear, thanks to the intuitive explanation provided by the authors. They also provide necessary background information on causal discovery and related literature.

# Significance

The problem that this paper aims to talks is significant in the field of tmeproal causal discovery. Although I don't think this paper fully addresses it, it made a reasonable progress towards the final destination.

**Weaknesses:**

# Weakness
Although the paper provide a solid progress towards the final destination, here are some limitations I am a bit worried about:

## Assumptions are too strong:
One assumption is that only 1 step window is considered in this paper. However, is this true in practice? Is it possible that some factors has long-standing effect (higher-order Markovian) to the target variable? Can your method handle such scenario?

For summary DAG, does it has to be a DAG? Since from general full-time DAG, the summary graph may not be a DAG and can contain cycles. Does your method handle such cases? For example, A(1) -> B(2) and B(1) -> A(2), and the summary graph is A<->B.

What are the consequence if some variables are not self-caused? That means the proxy variables may not exists. In that case, full identifiablity cannot be establish, right? You may want to add something like to what extend of identifiability your method can establish without such assumption.

## Comparison with baselines
The author includes some of the well-known baselines. Surprisingly, Dynotears performes reasonably ok considering that it is a linear model (in its original form). However, after Dynotears, several state-of-the-art baselines are proposed, like Rhino [1]. Can you compare your method with this nonlinear model?

[1]Gong, Wenbo, et al. "Rhino: Deep Causal Temporal Relationship Learning With History-dependent Noise." arXiv preprint arXiv:2210.14706 (2022). Code at https://github.com/microsoft/causica/tree/v0.0.0

## Scalability

How expensive it is to run such algorithm, since you need to test every pair. For real-world experiment, you scale it to 90 variables, how long does it need to run it? If your method can be scaled to higher-order SVAR, how does it scales with the window size?

**Questions:**

1. In figure 4 proxy model, do you mean $M(t_2)$?

**Limitations:**

The author did discuss its limitations but they only discuss the limitations of conditional independence test. Howver, in the weakness section, I have raised several improvements the author can consider. I suggest including these discussions in the limitations as well.

---

> ### Author Rebuttal · Authors · 2023-08-09
>
> Thank you for the highly constructive feedback and thoughtful suggestions on our paper. We address your concerns below.
>
> **About assumptions**:
>
> We first would like to point out that our assumptions, such as the first-order SVAR assumption and the self-causation assumption are commonly adopted in the literature [10-15,26-27].
>
> **Q1.** One assumption is that only 1 step window is considered in this paper. However, is this true in practice? Is it possible that some factors has long-standing effect (higher-order Markovian) to the target variable? Can your method handle such scenario?
>
> **A**: The first-order Markov model offers a practical approximation for numerous real-world situations. This includes scenarios like Reinforcement Learning tasks based on the Markovian Decision Process, neural activities in the brain [Valdes-Sosa et al. 2004], epidemic spreading [Anderson et al. 1991], and economic growth [Orcutt et al. 1969].
>
> The extension to the higher-order settings is not a trivial task, as longer causal chains may induce the bias in a much more complex way than shorter ones. We plan to address these cases in future work.
>
>
> [1] Valdes-Sosa PA. Spatio-temporal autoregressive models defined over brain manifolds. Neuroinformatics. 2004.
>
> [2] Cohen JE. Infectious diseases of humans: dynamics and control. JAMA. 1992.
>
> [3] Orcutt GH, Winokur Jr HS. First-order autoregression: inference, estimation, and prediction. Econometrica. 1969.
>
>
> **Q2.** For summary DAG, does it has to be a DAG? Since from general full-time DAG, the summary graph may not be a DAG and can contain cycles. Does your method handle such cases?
>
> **A**: Indeed, the summary graph can contain cycles. Our method can handle this case since it does not influence the identifiability results. We will modify the manuscript accordingly.
>
> **Q3.** What are the consequence if some variables are not self-caused? That means the proxy variables may not exists. In that case, full identifiablity cannot be establish, right? You may want to add something like to what extend of identifiability your method can establish without such assumption.
>
> **A**: Indeed, for this case, full identifiability cannot be established. What we can identify are ancestral equivalence classes [Plis et al. 2015]. We will supplement this discussion to the manuscript as suggested.
>
> [1] Plis S, Danks D, Freeman C, Calhoun V. Rate-agnostic (causal) structure learning. NeurIPS. 2015.
>
>
> **Comparison with Rhino**:
>
> Our method can outperform Rhino for about 30% in the f1-score on the synthetic dataset. Please refer to the supplementary PDF for details.
>
> **About scalability**:
>
> First note that we do NOT need to test every pair because most of the pairs can be screened out in step-3(a) with the necessary condition.
>
> It takes about 20 hours to run the real-world experiment. This time cost is acceptable considering that many of the baselines, e.g., [10,11,17], can not produce results in a feasible amount of time.
>
> To further validate the scalability of our method, we provide more experimental results in the supplementary PDF.
>
>
> **About Fig. 4**. Indeed, we mean $M(t_2)$. This is a typo.

---

> ### Comment · Reviewer_Y9xG · 2023-08-16
>
> Thanks for the authors' effort on addressing my concerns. They managed to addressed most of my concerns. But still, longer dependencies (longer than window 1) can still be present in practice, and this is a strong assumption in my personal opinions. I will keep my current score.

---

### Official Review · Reviewer_7zAn · 2023-07-04

**Soundness:** 3 good
**Presentation:** 3 good
**Contribution:** 2 fair
**Rating:** 5
**Confidence:** 5

**Summary:**

This paper proposed a non-parametric constraint-based algorithm that can identify the entire causal structure from subsampled data, which leverages the proxy variable to adjust the bias induced by the hidden variable.


**Strengths:**

- Concise and clear theoretical derivation. Introduce the proposed method by discussing the connections between different graphs step by step.


**Weaknesses:**

- The paper is rather incremental as the core of the method is based on [19] to use proxy variables to detect and eliminate the confounding effect brought by the subsampling.
- Moreover, the method proposed by [19] seems to require more assumptions but is not disclosed in this paper. And it is unclear whether those specific assumptions can be satisfied in this work.
- In addition, this work assumes the self-exciting property of the time-series which is not necessarily held and it is interesting to see how to find the proxy variables and what would be the result in this case.
- It could be better to provide more than 5 vertex numbers in the experiment.
- What would be the result if the causal graph becomes denser?


**Questions:**

See the weakness above.

**Limitations:**

Yes.

---

> ### Author Rebuttal · Authors · 2023-08-09
>
> Thank you for your efforts on our paper. We address your concerns below.
>
> **Q1.** The paper is rather incremental as the core of the method is based on [19] to use proxy variables to detect and eliminate the confounding effect brought by the subsampling.
>
> **A**: First note that our paper solves a very different problem from [19]. Our method is the first to achieve nonparametric causal identification in subsampled time series, setting it apart from existing methods that are limited to the linear case.
>
> Moreover, at the cores of our method lie the formulation of the subsampling bias (Def. 3.2, Prop. 3.3) and the identification of the separation set (Thm. 3.5 (1)). These analyses serve as the foundation for our use of proxy variables.
>
> **Q2.** Moreover, the method proposed by [19] seems to require more assumptions but is not disclosed in this paper. And it is unclear whether those specific assumptions can be satisfied in this work.
>
> **A**: [19] requires the smoothness of the structural equation and the invertibility of the transition matrix. All of them can be satisfied in our work.
>
> Due to space limits, we in Asm. 2.6 used the shorter version of their assumptions (Exam. 4.4, [19]). We will supplement the full assumptions to the manuscript as suggested.
>
> **Q3.** In addition, this work assumes the self-exciting property of the time-series which is not necessarily held and it is interesting to see how to find the proxy variables and what would be the result in this case.
>
> **A**: As stated in Rem. 2.11, the self-causation assumption is commonly used in time series. For cases where this assumption does not hold, no immediate guarantee of finding suitable proxies can be given. We will work on this in future work.
>
> **Q4.** It could be better to provide more than 5 vertex numbers in the experiment. What would be the result if the causal graph becomes denser?
>
> **A**: Our method is consistently accurate under different scales and densities. Please refer to the supplementary PDF for details.

---

### Official Review · Reviewer_XP2k · 2023-07-04

**Soundness:** 3 good
**Presentation:** 3 good
**Contribution:** 2 fair
**Rating:** 5
**Confidence:** 3

**Summary:**

In this paper, the author(s) propose a new technique to learn the summary graph of time-series data. As a motivation for their work, the author(s) discuss the interesting application of learning causal pathways in Alzheimer’s disease. The time series model studied in this work is quite general, and suitable for many applications. The author(s) provide a simple algorithmic approach to learn the summary graph. This algorithm is essentially based on verifying d-separation, by testing conditional independence. The author(s) conclude with an experimental comparison, showing that their method achieves superior performance than a baseline, using various synthetic datasets and a real-world medical application.

**Strengths:**

This paper studies an very important and difficult problem. The application of learning causal pathways in Alzheimer’s disease is very relevant. The paper is also well-written and easy to follow.

**Weaknesses:**

My main concern pertains the faithfulness assumption. Faithfulness it is useful for getting identifiability results, and it allows to recover the causal structure by testing conditional independence. However, in many scenarios there is no practical reason to assume faithfulness. The use of faithfulness significantly limits the novelty of their work. If I understand correctly, the implementation of their algorithm essentially uses conditional independence to recover the causal structure. Testing conditional independence is practically problematic. Hence, I also have doubts on the scalability of their method.

**Questions:**

Can you please describe exactly step 1 in your algorithm? How do you test conditional independence?

**Limitations:**

Yes, the author address limitations in Section 6.

---

> ### Author Rebuttal · Authors · 2023-08-09
>
> Thank you for your efforts on our paper. We address your concerns below and hope this can help you re-evaluate our paper.
>
> **Q1.** My main concern pertains to the faithfulness assumption. Faithfulness is useful for getting identifiability results, and it allows for recovery of the causal structure by testing conditional independence. However, in many scenarios, there is no practical reason to assume faithfulness. The use of faithfulness significantly limits the novelty of their work.
>
> **A**: Our method can work under the v-adjacency-faithfulness assumption [1], which requires that any two variables connected by an inducing path in the causal structure are dependent given any conditioning set. This assumption is strictly weaker than strong faithfulness and can handle common counterexamples such as the unshielded collider.
>
> The faithfulness assumption or its weaker variations have been widely utilized in constraint-based and score-based causal discovery approaches. This can be justified by the mathematical insignificance of unfaithful distributions (a lower-dimensional plane in a higher-dimensional space) [2-4]. Furthermore, a recent investigation [5] indicates that any causal learning method deemed effective must adhere to the standard practice of incorporating algorithms under the faithfulness condition. This highlights the essential nature and significance of the faithfulness condition.
>
> [1] Zhang J, Eberhardt F, Mayer W, Li MJ. ASP-based discovery of semi-Markovian causal models under weaker assumptions. IJCAI. 2019.
>
> [2] Meek C. Strong completeness and faithfulness in Bayesian networks. UAI. 1995.
>
> [3] Spirtes P, Glymour CN, Scheines R. Causation, prediction, and search. MIT press. 2000.
>
> [4] Uhler C, Raskutti G, Bühlmann P, Yu B. Geometry of the faithfulness assumption in causal inference. The Annals of Statistics. 2013.
>
> [5] Lin H, Zhang J. On learning causal structures from non-experimental data without any faithfulness assumption. Algorithmic Learning Theory. 2020.
>
>
> **Q2.** Testing conditional independence is practically problematic. Hence, I also have doubts on the scalability of their method.
>
> **A**: Our method is scalable to larger sizes of the causal graph. For instance, it takes only 3 hours to run a simulation experiment with 45 variables (see Fig. 1 in the supplementary PDF).
>
> In contrast, the baselines [10-12] (functional causal model-based), which use the EM algorithm for iterative estimation, cannot produce results in a feasible amount of time; The baseline Rhino [Gong et al. 2023] (score-based), which involves a double-nested optimization, requires up to 6-8 hours of training to converge.
>
> To summarize, we believe that scalability is a common challenge for causal discovery methods, not just constraint-based ones. Addressing this issue is beyond the scope of this paper.
>
> [1] Gong W, Jennings J, Zhang C, Pawlowski N. Rhino: Deep causal temporal relationship learning with history-dependent noise. ICLR. 2023.
>
> **Q3.** Can you please describe exactly step 1 in your algorithm? How do you test conditional independence?
>
> **A**: As mentioned in line 220, we use the Fast Causal Inference (FCI) algorithm implemented in the $\mathrm{causallearn}$ package to perform step 1. We use the kernel-based conditional independence test [Zhang et al. 2011].
>
> [1] Zhang K, Peters J, Janzing D, Schölkopf B. Kernel-based conditional independence test and application in causal discovery. UAI. 2011.

---

> > ### Comment · Reviewer_XP2k · 2023-08-15
> >
> > I would like to thank the reviewers for answering my questions. Given that the author(s) addressed my concerns in detail, and given the overall discussion and positive scores from the other reviewers, I will raise my score accordingly.

---

### Official Review · Reviewer_GTBD · 2023-07-15

**Soundness:** 2 fair
**Presentation:** 3 good
**Contribution:** 3 good
**Rating:** 6
**Confidence:** 3

**Summary:**

This paper studies the problem of subsampled time series in causal discovery, in which the unobserved time steps may lead to the existence of latent confounders. To this end, this paper proposes a constraint-based algorithm by leveraging proxy variables to remove the bias induced by latent confounders. The experimental results verify the effectiveness of the proposed algorithm.


**Strengths:**

1. This paper addresses the problem of subsampled time series in causal discovery, which is important but challenging.

2. The paper is well-structured and written.

3. The experimental results show that the proposed method outperforms several representative baselines.

**Weaknesses:**

1. In Theorem 2.8, extra assumptions are required for testing conditional independent relations in related literature, but they are not discussed in this paper.

2. How to search the proxy variable of target hidden variables. If an invalid proxy variable is selected, what is the output of the proposed algorithm?


3. In simulation data, the dimension of the random graph is only five. Can you show the performance of the proposed method in a larger-scale network?

**Questions:**

Refer to Weaknesses

**Limitations:**

Refer to Weaknesses

---

> ### Author Rebuttal · Authors · 2023-08-09
>
> Thank you for your efforts and valuable suggestions on our paper. We address your concerns below.
>
> **Q1.** In Theorem 2.8, extra assumptions are required for testing conditional independent relations in related literature, but they are not discussed in this paper.
>
> **A**: These assumptions are mentioned in Asm. 2.6 and Rem. 2.7.
>
>
> **Q2.** How to search the proxy variable of target hidden variables. If an invalid proxy variable is selected, what is the output of the proposed algorithm?
>
> **A**: The proxy of each hidden variable is itself at some observable time in the future, so no searching is needed. We mentioned this in Thm. 3.5 (line 190).
>
>
> **Q3.** In simulation data, the dimension of the random graph is only five. Can you show the performance of the proposed method in a larger-scale network?
>
> **A**: Our method is consistently accurate under different scales. Please refer to the supplementary PDF for details.

---

> > ### Comment · Reviewer_GTBD · 2023-08-12
> >
> > Thank you for your response. My score will remain unchanged.

---

### Author Rebuttal · Authors · 2023-08-09

We would like to express our gratitude to all the reviewers for their efforts and valuable comments. We are particularly pleased to hear that our work addresses an important and challenging problem (3AR7,Y9xG, GTBD, XP2K) and that our method is considered novel, solid (3AR7,Y9xG,7zAn), and well-presented (3AR7,Y9xG, GTBD,7zAn, XP2K).

Regarding the concerns raised, we first would like to emphasize that all our assumptions, including the faithfulness assumption, the SVAR assumption, and the self-causation (also known as autocorrelation) assumption, are commonly made in the literature [13-15, 26-27].

Furthermore, to demonstrate the scalability of our method, we conducted experiments on graphs with varying numbers of nodes $d=\{5,15,25,35,45\}$. The results are presented in Fig. 1 in the supplementary PDF. As shown, our algorithm consistently performs well ($F_1$-score = $\{0.94,0.94,0.93,0.91,0.90\}$) across different scales. These results validate that our method can effectively handle problems of various sizes.

---

### Decision · Program_Chairs · 2023-09-21

**Decision:**

Accept (poster)

**Comment:**

This paper proposed a constraint-based method that can infer the causal structures from subsample time series data without any parametric constraints. Specifically, this method leverages the proxy variable to adjust the bias induced by the hidden variable. The idea is considered novel (reviewers GTBD, XP2k, Y9xG, 3AR7), and the method is sound and solid (reviewers 7zAn, Y9xG, 3AR7).
Reviewers proposed issues about assumption (reviewers XP2k, Y9xG, 7zAn, GTBD), implementation (reviewers XP2k), and experimental results (reviewers GTBD, Y9xG, 7zAn). Fortunately, the authors have addressed the main issues proposed by the reviewers (XP2k, 7zAn, GTBD, 3AR7). The decision is acceptance.